# Photon momentum transfer and partitioning: from one to many

Xiaodan Mao [1], Hongcheng Ni [1,2] ✉, Kang Lin [3,4] ✉, Pei-Lun He [5,6,7], Hao Liang [8], Sebastian Eckart [9], Feng He [6,7], Kiyoshi Ueda [1,10], Reinhard Dörner [9] & Jian Wu [1,2,11] ✉

The transfer of photon momentum is indispensable in initiating and directing light-matter interactions, which underpins a plethora of fundamental physical processes from laser cooling to laser particle acceleration. The transferred photon momentum is distributed between the photoelectron and the residual ion upon ionization. Our study presents a general and consistent framework for photon momentum transfer covering an arbitrary number of absorbed photons. Our results bridge the gap between the previously considered limiting cases of single-photon and multi-photon strong-field ionization and suggest revising the current consensus for the multi-photon limit by demonstrating that with each additional photon absorbed above the ionization threshold, the photoelectron acquires on average twice the momentum of the absorbed photon. Our work paves the pathway towards a comprehensive understanding of the fundamental processes of photon momentum transfer in light-matter interactions, with implications for both theoretical physics and practical applications that harness the transfer of photon momentum.

The universal exchange of photon energy and momentum triggers and steers light-matter interactions, resulting in a plethora of intricate physical phenomena. Each photon carries an energy of $\hbar\omega$ and a momentum of $\hbar\omega/c$, with $\omega$ the angular frequency of light and $c$ the light speed in vacuum [atomic units (a.u.) are used throughout unless stated otherwise, where the electron mass $m_e$, elementary charge $e$, and reduced Planck constant $\hbar$ are set to 1 a.u.]. Upon photoionization, the absorbed photon energy and momentum are distributed among the resulting photoelectrons and ions. While the linear momentum of a photon is typically very small and often disregarded under the dipole approximation, there are numerous fundamental physical processes that are driven by photon momentum, such as radiation pressure, laser cooling[1], second-harmonic generation[2], Compton scattering[3], and the Kapitza–Dirac effect[4]. Understanding the partitioning of the transferred photon momentum among the photoelectrons and ions during light-matter interactions is therefore a critical question to address (Fig. 1).

With recent advances in laser and detection technologies, the subtle transfer of photon momentum can now be measured with unprecedented precision. Remarkably, it is found that the distribution of the photon momentum between the photoelectron and ion significantly depends on the specific process at play. In 1930, it has already been predicted that, in the high-energy limit of single-photon ionization of an $s$-shell, the photoelectron acquires a linear momentum

[1]State Key Laboratory of Precision Spectroscopy, East China Normal University, Shanghai, China. [2]Collaborative Innovation Center of Extreme Optics, Shanxi University, Taiyuan, Shanxi, China. [3]School of Physics, Zhejiang University, Hangzhou, China. [4]Zhejiang Key Laboratory of Micro-nano Quantum Chips and Quantum Control, Zhejiang University, Hangzhou, China. [5]Max-Planck-Institut für Kernphysik, Heidelberg, Germany. [6]Key Laboratory for Laser Plasmas (Ministry of Education), School of Physics and Astronomy, Shanghai Jiao Tong University, Shanghai, China. [7]Collaborative Innovation Center for IFSA (CICIFSA), Shanghai Jiao Tong University, Shanghai, China. [8]Max-Planck-Institut für Physik komplexer Systeme, Dresden, Germany. [9]Institut für Kernphysik, Goethe-Universität Frankfurt, Frankfurt am Main, Germany. [10]Department of Chemistry, Tohoku University, Sendai, Japan. [11]Chongqing Key Laboratory of Precision Optics, Chongqing Institute of East China Normal University, Chongqing, China. ✉e-mail: hcni@lps.ecnu.edu.cn; klin@zju.edu.cn; jwu@phy.ecnu.edu.cn

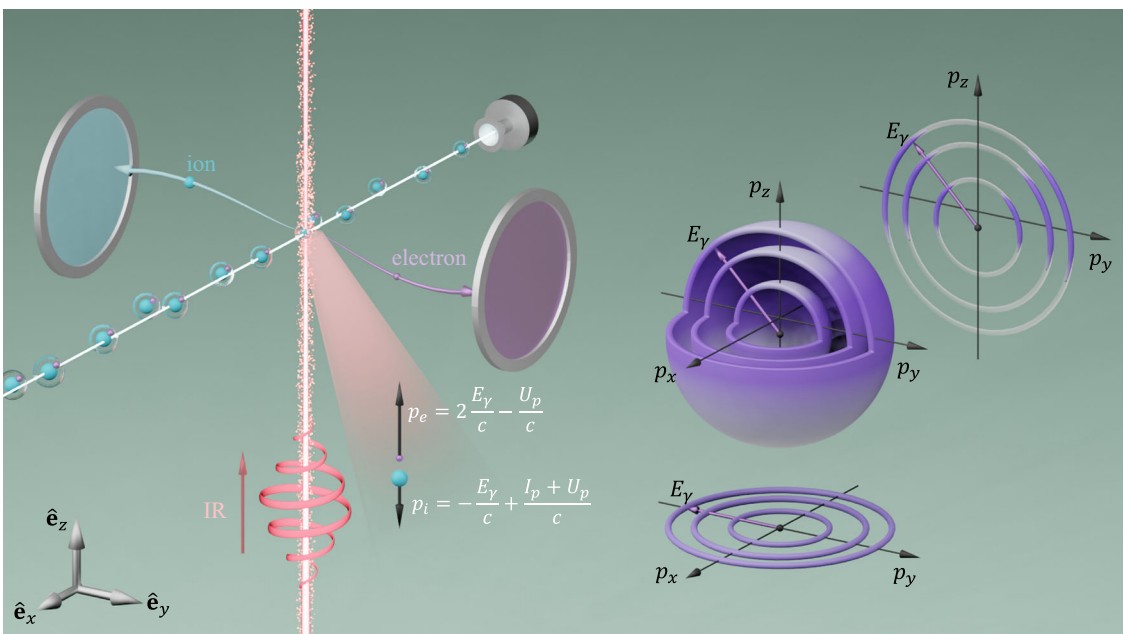

**Fig. 1 | Sketch of photon momentum transfer and partitioning.** A circular laser pulse ionizes Xe atoms, transferring on average a linear momentum of $p_e = 2E_\gamma/c - U_p/c$ to photoelectrons and $p_i = -E_\gamma/c + (I_p + U_p)/c$ to ions along the light propagation direction $z$, where $U_p$ is the ponderomotive potential, $I_p$ is the ionization potential, and $c$ is the light speed in vacuum. The photoelectrons and ions are detected coincidently by two detectors at the opposite sides along the $y$ direction. Photoabsorption from the light pulse gives rise to ATI peaks in the photoelectron momentum distribution (PMD), as indicated by the purple concentric spherical shells, which have a common shift of $-U_p/c$ along the light propagation direction $z$. These concentric shells suggest a definition of the dressed energy $E_\gamma \equiv \frac{1}{2m_e}[p_\perp^2 + (p_z + U_p/c)^2]$, where $p_z$ is the momentum in the $z$ direction and $p_\perp^2 = p_x^2 + p_y^2$ is that in the polarization plane. The concentric rings at the bottom and the side represent the cut of the PMD at $p_z = 0$ and $p_x = 0$, respectively. The PMD tilts in the forward direction, as indicated by the color distribution along the rings in the side cut, such that photoelectrons gains on average a positive linear momentum.

of $\frac{8}{5}\frac{E}{c}$[5-9], with $E$ the photoelectron energy. This prediction has been verified experimentally recently[10,11]. In the multi-photon limit of tunneling ionization, in contrast, the photoelectron acquires a linear momentum of about $\frac{E}{c} + \frac{I_p}{3c}$[12-23], with $I_p$ the ionization potential. This leads to the question: What governs the dependence of photon momentum transfer on ionization dynamics? More specifically, why does the slope in front of $\frac{E}{c}$ vary with the number of photons absorbed from the driving light field? And how to bridge the single-photon and multi-photon limits? To answer these questions, in this work we establish a general framework for photon momentum transfer applicable to an arbitrary number of photons involved.

Above-threshold ionization (ATI)[24-26] provides a unique platform to explore the distribution of the transferred photon energy and momentum in a single experiment. In order to avoid backscattering close to the multi-photon limit, we utilize circularly polarized light pulses. Given that the photoion is significantly heavier than the photoelectron, the latter typically retains most of the photon energy. Consequently, the energy of the photoelectron can be expressed as $E = n\hbar\omega - I_p - U_p'$, where $n \in \mathbb{N}$ is the number of photons absorbed, and $U_p' = [1 + p_z/(m_e c)]U_p$ is the ponderomotive potential beyond the dipole approximation[17,27-29], with $U_p$ its dipole counterpart and $p_z$ the photoelectron momentum along the laser propagation direction $z$. The ponderomotive energy $U_p$ represents the average quiver energy of a free electron in a laser field. In a long circular laser field, $U_p = e^2 A_0^2/(2m_e) = e^2 F_0^2/(2m_e\omega^2)$ remains constant, with $A_0$ and $F_0$ being the amplitudes of the vector potential and the electric field, respectively. Clearly, the photoelectron energy spectrum exhibits distinct peaks, known as ATI peaks, separated by one photon energy, in accordance with the shells or rings present in the photoelectron momentum distribution (PMD, see Fig. 1). Additionally, by reshaping $E = n\hbar\omega - I_p - U_p'$ up to order $1/c$, it is easy to show that all ATI momentum rings are shifted by a common value of $-U_p/c$ along the laser propagation direction $z$:

$$E_\gamma \equiv \frac{1}{2m_e}\left[p_\perp^2 + \left(p_z + \frac{U_p}{c}\right)^2\right] = n\hbar\omega - I_p - U_p, \quad (1)$$

where $p_\perp^2 = (p_x^2 + p_y^2)$ is the transverse momentum in the polarization plane and a transverse energy $E_\perp = p_\perp^2/(2m_e)$ can be defined accordingly. Equation (1) motivates a definition of the "dressed energy" $E_\gamma$, which is a concept that will enable us to reveal the general rule of the photon momentum transfer. Based on this, we establish a consistent framework for photon momentum transfer covering an arbitrary number of absorbed photons. In addition, we challenge the existing knowledge in the multi-photon limit by showing that the slope is 2 instead of 1 for multi-photon ionization. In other words, with each additional photon absorbed, the photoelectron acquires a momentum that is twice that of a single photon, and the ion gains a recoil momentum anti-parallel to the momentum of the absorbed photon, as schematically shown in Fig. 1.

## Results
### Photon momentum transfer in above-threshold ionization
Experimentally, we carry out studies using a cold-target recoil-ion momentum spectroscopy (COLTRIMS) setup[30]. Careful calibration of the zero point in the momentum spectrum, which is crucial for the present study, has been enabled via the formation of a standing wave in the vacuum chamber of the COLTRIMS reaction microscope by using two counter-propagating laser pulses. During the measurement of linear momentum transfer, one of the two counter-propagating lasers is turned off. The coincident detection capability of the COLTRIMS setup enables a precise recording of the PMD. See the Methods section for details.

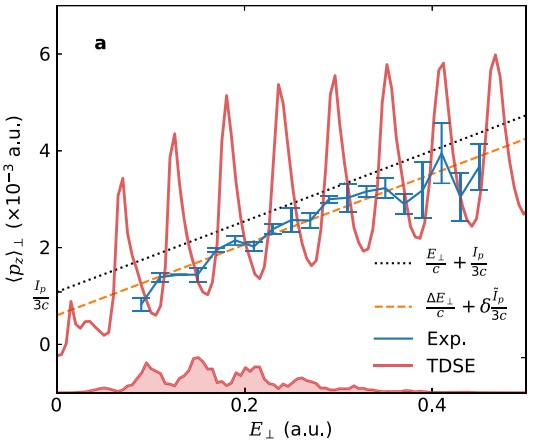
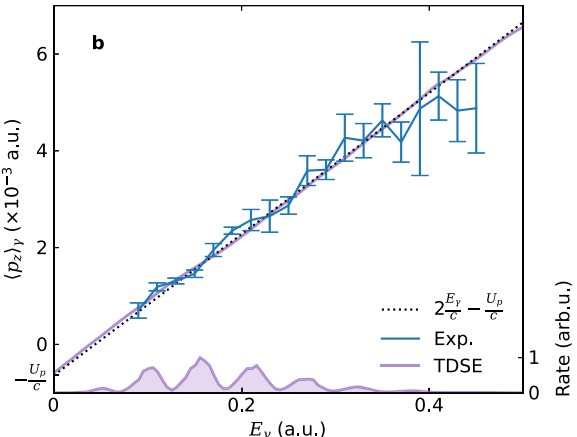

**Fig. 2 | ATI spectrum and linear momentum transfer. a** Linear momentum transfer $\langle p_z \rangle_\perp$ as a function of the transverse energy $E_\perp$. **b** Linear momentum transfer $\langle p_z \rangle_\gamma$ as a function of the dressed energy $E_\gamma$ [Eq. (1)]. The shaded areas stand for the ATI spectra. The red and purple lines are calculated by TDSE, and the blue lines denote experimental results. The binning size employed is 0.02 a.u., and the results are robust against variations in binning size. In (**a**), the black dotted line represents $\langle p_z \rangle_\perp = E_\perp/c + I_p/(3c)$ [Eq. (2)] and the orange dashed line denotes $\langle p_z \rangle_\perp = \Delta E_\perp/c + \delta \tilde{I}_p/(3c)$ [Eq. (3)]. In (**b**), the black dotted line represents $\langle p_z \rangle_\gamma = 2E_\gamma/c - U_p/c$ [Eq. (4)].

Figure 2 displays the measured photon momentum transferred to the photoelectron, $\langle p_z \rangle_\perp$, for the Xe atom as a function of the transverse energy $E_\perp$ in panel (a) and $\langle p_z \rangle_\gamma$ as a function of the dressed energy $E_\gamma$ in panel (b). While $\langle p_z \rangle_\perp^{(\mathrm{Exp.})}$ shows a qualitative agreement with[9,23]

$$\langle p_z \rangle_\perp = \frac{E_\perp}{c} + \frac{I_p}{3c}, \tag{2}$$

or a better agreement with[20]

$$\langle p_z \rangle_\perp = \frac{\Delta E_\perp}{c} + \delta \frac{\tilde{I}_p}{3c} = \frac{E_\perp - E_{\perp 0}}{c} + \delta \frac{I_p + E_{\perp 0}}{3c}, \tag{3}$$

where $\langle \cdot \rangle_\perp$ denotes integration along the iso-transverse-energy-$E_\perp$ surface, $E_{\perp 0}$ denotes the initial kinetic energy at the tunnel exit, $\tilde{I}_p$ represents the effective ionization potential, and $\delta$ originates from the prefactor in the ionization rate (see Supplementary Materials for details); $\langle p_z \rangle_\gamma^{(\mathrm{Exp.})}$, on the other hand, shows quantitative agreement with

$$\langle p_z \rangle_\gamma = 2\frac{E_\gamma}{c} - \frac{U_p}{c}, \tag{4}$$

where $\langle \cdot \rangle_\gamma$ represents integration along the iso-dressed-energy-$E_\gamma$ surface. Clearly, the slopes of the average linear momentum transfer differ by a factor of 2 with respect to their respective energy.

The comparison above demonstrates that, the observed photon momentum transfer depends not only on the particular ionization mechanism at play but also on different ways to extract its average value. In previous experimental studies, the average photon momentum transfer has been obtained by integration over the PMD along the $p_z$ axis, leading to the approximate relation $\langle p_z \rangle_\perp = \frac{E_\perp}{c} + \frac{I_p}{3c}$ [Eq. (2)], see Fig. 2a. Notably, the transverse energy $E_\perp = p_\perp^2/(2m_e) = (p_x^2 + p_y^2)/(2m_e)$ lacks the $p_z$ component along the laser propagation direction and is thus not the full photoelectron energy. In contrast, upon photoabsorption, the full photon energy $\omega$ is deposited to the photoelectron and is reflected in the increase in its full energy. Therefore, $E_\perp/c$ does not mirror the photon momentum and Eq. (2) does not answer the question how the photon momentum is distributed between the photoelectron and the ion.

In order to pinpoint the origin of the vastly different conclusions of linear momentum transfer from different perspectives, we numerically solve the time-dependent Schrödinger equation (TDSE) for the Xe atom interacting with a circularly polarized laser pulse. See the Methods section for details. Figure 3a shows the calculated PMD across the $p_\perp p_z$ plane, where the angular direction in the polarization plane ($xy$ plane) has been integrated over. The overall PMD is centered around $p_\perp \approx A_0 = 0.42$ a.u., a well-known conclusion of strong-field ionization closely related to the attoclock principle[31,32]. In addition, the concentric spherical rings arise as ATI peaks.

Scrutiny into regions surrounding different ATI peaks uncovers how the linear momentum transfer appears differently from different perspectives. Near the second ATI order $N = 2$, the distribution of linear momentum is obtained by integration along either red (iso-$E_\perp$) or purple (iso-$E_\gamma$) solid lines in Fig. 3a, leading to Fig. 3b. To enhance the visibility, the widths of the distributions have been reduced by a factor of 100. Clearly, $\langle p_z \rangle_\perp > \langle p_z \rangle_\gamma$ here near $N = 2$. Near the fifth order $N = 5$, the distribution of linear momentum is obtained similarly, resulting in Fig. 3c. Evidently, it is in stark contrast that $\langle p_z \rangle_\perp < \langle p_z \rangle_\gamma$ near $N = 5$, where the order has been reversed. Therefore, as the photoelectron energy increases, the increase in $\langle p_z \rangle_\gamma$ is greater than that of $\langle p_z \rangle_\perp$, leading to different slopes as illustrated in Fig. 2.

It is clear from Fig. 3 that $\langle p_z \rangle_\perp$ obtained from iso-$E_\perp$ integration mixes different ATI orders. This leads to oscillations of $\langle p_z \rangle_\perp$ as a function of the transverse energy $E_\perp$, shown as the red solid line in Fig. 2a. This oscillatory behavior becomes smoother after focal volume averaging (see Supplementary Materials for details). In contrast, the energy-domain $\langle p_z \rangle_\gamma$ extracted from iso-$E_\gamma$ integration, shown as the purple solid line in Fig. 2b, is very smooth and aligns very well with Eq. (4).

Figure 2 shows additionally the distributions of the transverse energy $E_\perp$ and the dressed energy $E_\gamma$ as shaded areas. Evidently, the distribution of the dressed energy $E_\gamma$ has smoother and more distinct peaks than that of the transverse energy $E_\perp$, because the former does not mix different ATI peaks. We have also checked that focal volume averaging, albeit leading to smeared-out ATI peaks, does not hamper our conclusions. See the Supplementary Materials for details.

## Photon-number-resolved linear momentum transfer

To analyze the photon momentum transfer in more detail, we employ the nondipole strong-field approximation (ndSFA), in which, compared to the full numerical solution using TDSE, the Coulomb interaction in the final state is neglected, simplifying numerical evaluation and theoretical analysis. A direct evaluation of the ndSFA transition amplitude will give the momentum distribution, from which the

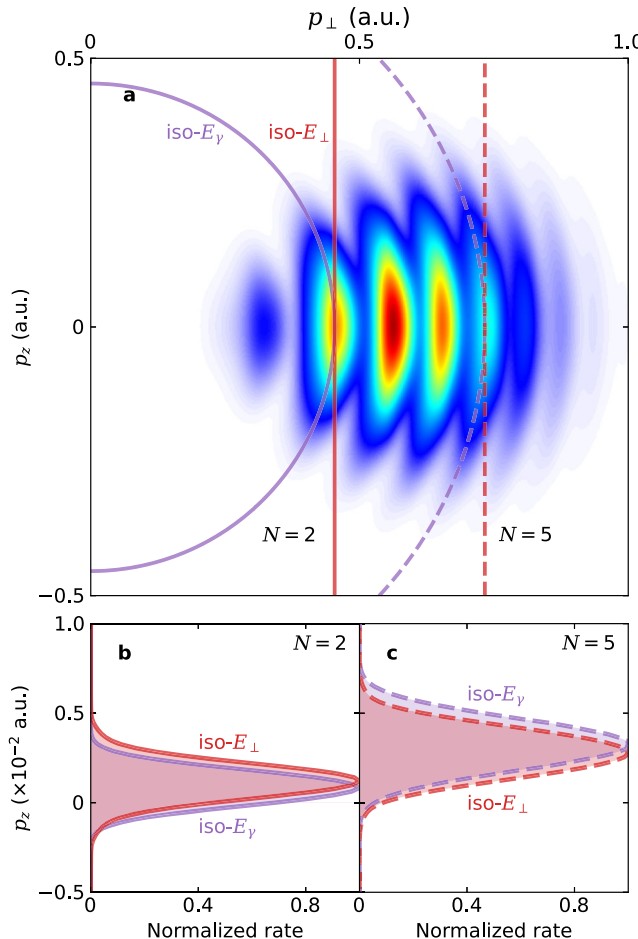

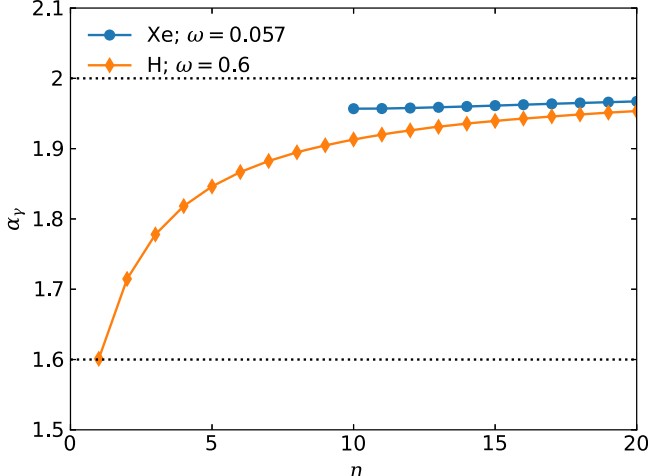

**Fig. 4 | Slope parameter $\alpha_\gamma$ as a function of the number of absorbed photons $n$.** for the ATI of the Xe atom (blue line with circles) and the H atom (orange line with diamonds).

**Fig. 3 | Calculated photoelectron momentum distribution (PMD). a** PMD calculated by TDSE, where red and purple lines represent iso-transverse-energy-$E_\perp$ and iso-dressed-energy-$E_\gamma$ surfaces, respectively, and solid and dashed lines denotes the second and fifth ATI peaks, respectively. **b** linear momentum transfer collected along iso-transverse-energy-$E_\perp$ (red line) and iso-dressed-energy-$E_\gamma$ (purple line) surfaces for the second ATI peak. **c** Same as (**b**), but for the fifth ATI peak. For better visibility, the $p_z$ distributions in (**b**) and (**c**) have been squeezed by a factor of 100 without changing their peak value.

resolved linear momentum transfer to the photoelectron. See the Methods section for derivations.

For the limiting case of single-photon ionization, $n = 1$ and $U_p$ is typically very small and can be neglected. In this scenario, $\langle p_z \rangle_\gamma$ reduces to $8E/5c$ for hydrogen[5–11]. For the other limiting case of tunneling ionization, where the involved photon number $n$ is large, $\langle p_z \rangle_\gamma$ reduces to $2E_\gamma/c - U_p/c$, thereby reproducing Eq. (4). In the present scenario of ATI, the minimal number of photons needed to produce ionization is $n_0 = \text{ceil}[(I_p + U_p)/(\hbar\omega)] = 10$, fulfilling the large-$n$ requirement to make Eq. (4) a valid approximation.

We define the slope parameter $\alpha_\gamma = 4\{n + \nu/[1 - U_p/(n\hbar\omega)]\}/(2n + 3)$ to quantify the slope of linear momentum transfer as a function of the number of absorbed photons, which is shown in Fig. 4. The blue line with circles is obtained for Xe interacting with an 800 nm laser at a peak intensity of $4 \times 10^{13}$ W/cm². Evidently, the slope parameter $\alpha_\gamma$ is close to 2 within the range shown. The orange line with diamonds corresponds to the case of the H atom interacting with a laser at the same intensity and an angular frequency of $\omega = 0.6$ a.u.. Obviously, $\alpha_\gamma = 1.6$ when $n = 1$, and it approaches 2 as the number of absorbed photons $n$ increases.

## Field dressing and photon momentum partitioning

At last, we would like to elucidate the rationale behind employing the "dressed energy" $E_\gamma$ as a means to quantify the partitioning of photon momentum transfer. Equation (1) suggests that both the photoelectron energy and linear momentum in the ATI process are simultaneously influenced by the laser dressing, manifested as the $U_p$ term. At low laser intensity, $U_p$ in Eq. (1) is nearly negligible, and the photoionization process is mainly influenced by the frequency of the laser field. In contrast, under a strong laser field, the photoelectron returns an energy $U_p$ to the field, with a concurrent return of linear momentum $U_p/c$ to the field. The energy return of $U_p$ is evident in the last term of Eq. (1), while the linear momentum return of $U_p/c$ is observed as a common shift to the center of all ATI momentum rings. Not surprisingly, these energy and linear momentum exchanges adhere to the photon's dispersion relation. Within the dipole approximation, it is valid to consider only the influence of the laser field on the energy. However, in the nondipole regime, we must simultaneously analyze the consistent adjustments to both the photoelectron energy and linear momentum. Therefore, it is helpful to define a "dressed energy" $E_\gamma$ as given in Eq. (1), which represents the center-shifted full photoelectron energy. The virtue of $E_\gamma$ is that, in the nondipole case, this

energy-resolved linear momentum transfer can be extracted. In the multi-photon limit of tunneling ionization, the linear momentum transfer can be obtained by applying an additional saddle-point approximation to ndSFA, which we term ndSPA. There, Eqs. (2)–(4) can be derived fully analytically. See the Methods section for details.

To gain further insights into the physical origin of the photon-number-dependent linear momentum transfer, we apply the long-pulse approximation to ndSFA, leading to the nondipole Keldysh–Faisal–Reiss model (ndKFR) in a photon-number-resolved manner. Thereby, we obtain the photon-number-resolved linear momentum transfer as a key finding of our current work:

$$\langle p_z \rangle_\gamma = \frac{4\left(n + \frac{\nu}{1 - \frac{U_p}{n\hbar\omega}}\right)}{2n + 3} \frac{E_\gamma}{c} - \frac{U_p}{c}, \qquad (5)$$

where the dimensionless parameter $\nu = Z/\sqrt{2I_p/[2I_p^{(H)}]}$, with the ionization energy $I_p = 0.4457$ a.u., the ionization energy of the hydrogen atom $I_p^{(H)} = 0.5$ a.u., and the asymptotic charge $Z = 1$ for the valence shell of Xe. We note that Eq. (5) is valid for an arbitrary number of photons absorbed and can act as a general framework for photon-number-

quantity is quantized analogously to the the quantization of the photoelectron energy in the dipole case.

We consider now the partitioning of the linear momentum between the photoelectron and the ion. Under the influence of the laser field, a total energy of $n\hbar\omega - U_p = E_\gamma + I_p$ is deposited to the target, and a corresponding linear momentum of $(n\hbar\omega - U_p)/c = (E_\gamma + I_p)/c$ is transferred to the center of mass of the photoelectron and the ion. Both experimental and theoretical analyses have verified that an average linear momentum of $2E_\gamma/c - U_p/c$ is transferred to the photoelectron. Thus, by momentum conservation, the residual linear momentum of $-E_\gamma/c + (I_p + U_p)/c$ is transferred to the ion, as illustrated in Fig. 1.

## Discussion

We have established a general and consistent framework for photon linear momentum transfer applicable to an arbitrary number of photons absorbed that bridges the single-photon and multi-photon limits, using the prototypical process of above-threshold ionization. For the multi-photon limit, our results suggest that the established understanding needs to be reconsidered, as we conclusively show that, for each additional photon absorbed above the threshold, the photoelectron acquires on average twice of the photon momentum, while the photoion acquires a recoil linear momentum opposite to the light propagation direction. Our findings underscore a perspective of light-matter interaction involving both energy absorption and momentum transfer.

Distinguishing our approach from preceding studies, our research introduces a consistent framework for photon momentum transfer that takes into account the entirety of the photoelectron's energy. This comprehensive consideration of energy stands in contrast to earlier analyses, which focused on the partial transverse energy of the electron that does not account for the full photon momentum. Our approach, therefore, offers an unprecedented understanding of how photon momentum is partitioned during light-matter interactions that cover the range from single-photon absorption, to the multi-photon and the tunneling regime. This improved understanding will pave the way for further work that harnesses the transfer of photon momentum and further research on light-matter interactions.

## Methods

### Experimental setup

Experimentally, we adopt the same strategy as described in refs. 29,33–36. We only give a brief introduction here. We obtain two laser beam pathways of opposite propagation directions by splitting the output (25 fs, 800 nm, 10 kHz) of a Ti:Sapphire laser system (Coherent Legend Elite) using a 50% dielectric beam splitter. In both pathways, a neutral filter and two waveplates of half and quarter wavelengths are used to adjust the intensity and polarization. For this experiment, the two beams are shaped to circular polarizations, followed by focusing into the vacuum chamber of a COLTRIMS reaction microscope[30] from two opposite sides using two independent lenses ($f = 25$ cm) onto the same spot inside a supersonic gas jet of Xe atoms. To minimize the systematic errors during the data collection, two motorized shutters are used to toggle between the two pathways every 3 minutes. This procedure allows us to eliminate most systematical errors since the expected changes of the linear momentum transfer are on the order of 0.001 a.u.

To ensure circular polarization of the laser fields entering the vacuum chamber, we carefully controlled the polarization state using a combination of a half-wave plate and a quarter-wave plate in both pathways. This setup effectively compensates the differences in reflectivity between the s- and p-polarization components of the circularly polarized light. The degree of circular polarization achieved was estimated to be better than 0.9, approaching the ideal value of 1. The peak intensity in the laser focus is estimated to be $4 \times 10^{13}$ W/cm² with an uncertainty of ±20%. The laser intensity has been calibrated from the most probable radial momentum and scaled according to the power measured by a power meter.

The electrons and ions created from single ionization of Xe atoms are guided to two time- and position-sensitive detectors at opposite ends of the spectrometer by applying a static electric field of 29.8 V/cm. The spectrometer consists of acceleration and field-free drift regions for both ions and electrons to realize high momentum resolution. For the electron side, the acceleration and drift regions are 15 and 30 cm, respectively. For the ion side, the acceleration and drift regions are 58 and 108 cm, respectively. We did not employ a magnetic field to guide electrons and ions. Instead, we utilized a $\mu$-metal shield within the vacuum chamber to effectively screen external magnetic fields, particularly the Earth's magnetic field. The electron (ion) detector is composed of a three-layer (two-layer) stack of multichannel plates followed by a three-layer delay-line anode[37].

From the measured times-of-flight and positions-of-impact, the three-dimensional momenta of the electrons and ions are retrieved coincidently. The $y$-direction is the time-of-flight direction of the COLTRIMS reaction microscope. We employed coincident detection of the Xe ion (Xe⁺) and the electron (e⁻). By applying a momentum gate along the polarization direction, we ensured that only the ion and electron pairs originating from the same parent atom were selected, based on momentum conservation. With the above configuration, the single-event momentum resolution for a single electron is 0.003 a.u. in $p_x$ and $p_z$ directions and 0.03 a.u. in the $p_y$ direction.

To retrieve the expected linear momentum transfer, an integration over an adequate range is necessary. The present experimental setup, however, is limited in the recording range of $p_z$ due to the long spectrometer length and small size of the electron detector. Therefore, the peak value of the linear momentum transfer has been used as a replacement for the expectation value. We show that the peak and expectation values are essentially identical. See the Supplementary Materials for details.

The error bars in Fig. 2 represent statistical uncertainties associated with the experimental events. Given that our analysis involves the use of a standing wave, and the final results are derived by subtracting the forward and backward curves, any uncertainties related to the spectrometer calibration effectively cancel out along the light-propagation direction. Consequently, the primary source of uncertainty in our results is statistical error, stemming from the inherent variability of the experimental measurements.

### Time-dependent Schrödinger equation

The strong-field ionization of Xe in a circularly polarized laser field is simulated by numerically solving the three-dimensional TDSE under the single-active-electron approximation with the minimal-coupling nondipole Hamiltonian

$$H = \frac{1}{2m_e}[\mathbf{p} + e\mathbf{A}(\eta)]^2 + V(\mathbf{r}), \qquad (6)$$

where the atomic potential $V(\mathbf{r}) = -1/\sqrt{r^2 + a_0}$ with $a_0$ the soft-core parameter, and the vector potential of the laser pulse $\mathbf{A}(\eta)$ is expressed as

$$\mathbf{A}(\eta) = A_0 \cos^4(\omega\eta/2N_L)\left[\cos(\omega\eta)\hat{\mathbf{e}}_x + \sin(\omega\eta)\hat{\mathbf{e}}_y\right], \qquad (7)$$

polarized across the $xy$ plane, where $\eta = t - z/c$ is the light-cone time, the angular frequency $\omega$ corresponds to a central wavelength of 800 nm, the amplitude $A_0$ corresponds to a peak laser intensity of $4 \times 10^{13}$ W/cm², and $N_L = 10$ is the number of optical cycles. The corresponding electric field is $\mathbf{F}(\eta) = -\partial_t\mathbf{A}(\eta)$.

The vector potential $\mathbf{A}(\eta)$ can be approximated up to order $1/c$ as

$$\mathbf{A}(\eta) = \mathbf{A}\left(t - \frac{z}{c}\right) = \mathbf{A}(t) + \frac{z}{c}\mathbf{F}(t), \qquad (8)$$

where $\mathbf{A}(t)$ and $\mathbf{F}(t)$ represent the vector potential and electric field evaluated at the nucleus $z = 0$, respectively. The nondipole Hamiltonian can thus be rewritten up to order $1/c$ as

$$H = \frac{1}{2m_e}[\mathbf{p} + e\mathbf{A}(t)]^2 + \frac{e}{m_e}\frac{z}{c}[\mathbf{p} + e\mathbf{A}(t)] \cdot \mathbf{F}(t) + V(\mathbf{r}). \tag{9}$$

Performing a unitary transformation, we may obtain the expression for the transformed nondipole Hamiltonian[20,38]

$$H = \frac{1}{2m_e}\left\{\mathbf{p} + e\mathbf{A}(t) + \frac{\hat{\mathbf{e}}_z}{c}\left[\frac{e}{m_e}\mathbf{p} \cdot \mathbf{A}(t) + \frac{e^2}{2m_e}A^2\right]\right\}^2 + V\left[\mathbf{r} - \frac{e}{m_e}\frac{z}{c}\mathbf{A}(t)\right]. \tag{10}$$

In this transformed Hamiltonian [Eq. (10)], the spatial part and momentum part are separated, enabling straightforward application of the split-operator Fourier method for the solution of the TDSE. It is solved on a spatial grid with 1024 points and a spatial step size of $\Delta x = 0.3$ in each of the three dimensions. The time step is $\Delta t = 0.02$. Setting the soft-core parameter $a_0 = 0.03159$, the ground state is obtained using the imaginary-time propagation method with an energy $E_0 = -I_p = -0.4457$ a.u., matching the experimental value of the first ionization potential of Xe. Note that we have used the ground $s$ state of the present potential $V(\mathbf{r})$ to represent the valance shell of Xe. For the multi-photon limit of ionization, the specific structure of the initial state plays a negligible role.

The ground-state wave function is then evolved under the laser field using the nondipole Hamiltonian [Eq. (10)]. The outgoing wave function is damped by an absorbing boundary of the form $1/[1 + \exp\{(r - r_0)/d\}]$ with $r_0 = 138.6$ and $d = 4$ to avoid reflection from the grid border. The absorbed wave function at each time step is cumulatively projected onto nondipole Volkov states[39,40]

$$|\psi_\mathbf{p}(t)\rangle = |\mathbf{p}\rangle \exp\left\{-\frac{i}{\hbar}\int^t d\tau \frac{1}{2m_e}\left\{\mathbf{p} + e\mathbf{A}(\tau) + \frac{\hat{\mathbf{e}}_z}{c}\left[\frac{e}{m_e}\mathbf{p} \cdot \mathbf{A}(\tau) + \frac{e^2}{2m_e}A^2(\tau)\right]\right\}^2\right\} \tag{11}$$

to obtain the PMD $W(\mathbf{p})$. Finally, the expectation value of linear momentum transfer can be calculated as

$$\langle p_z\rangle = \frac{\int p_z W(\mathbf{p})d\mathbf{p}}{\int W(\mathbf{p})d\mathbf{p}}. \tag{12}$$

**Nondipole strong-field approximation**

The ndSFA neglects the Coulomb interaction in the final state as well as the laser polarization in the initial state, and the corresponding transition amplitude in the length gauge can be written as[39]

$$M_{\text{ndSFA}}(\mathbf{p}) = -\frac{i}{\hbar}\int_{-\infty}^{\infty} d\eta\left[e\left(1 - \frac{p_z}{m_e c}\right)i\nabla_\mathbf{k}\psi_0(\mathbf{k}) \cdot \mathbf{F}(\eta)e^{\frac{i}{\hbar}S(\eta)}\right], \tag{13}$$

where $\psi_0(\mathbf{k})$ is the ground-state wave function in the momentum representation evaluated at $\mathbf{k} = \mathbf{p} + e\mathbf{A}(\eta) - (E + I_p)\hat{\mathbf{e}}_z/c$ with $E = p^2/(2m_e) = (p_\perp^2 + p_z^2)/(2m_e)$, and the nondipole action is expressed as $S(\eta) = \int^\eta dt\{E + [(e/m_e)\mathbf{p} \cdot \mathbf{A}(t) + e^2A^2(t)/(2m_e)]/[1 - p_z/(m_e c)] + I_p\}$. Direct evaluation of Eq. (13) will give the momentum distribution, from which the energy-resolved linear momentum transfer can be extracted.

**Nondipole saddle-point approximation**

Further applying the nondipole saddle-point approximation with nonadiabatic expansion (ndSPA) on top of ndSFA, the transition rate

can be written, up to exponential accuracy, as[20,41,42]

$$W_{\text{ndSPA}}(\mathbf{p})$$
$$= \exp\left\{-\frac{2\left[2I_p + \frac{1}{m_e}p^2 + \left(1 + \frac{p_z}{m_e c}\right)\left(\frac{e}{m_e}2\mathbf{p} \cdot \mathbf{A} + \frac{e^2}{m_e}A^2\right)\right]^{3/2}}{3\hbar\sqrt{\left(1 + \frac{p_z}{m_e c}\right)\frac{e}{m_e}(eF^2 - \mathbf{v}_\perp \cdot \dot{\mathbf{F}})}}\right\}$$
$$= \exp\left\{-\frac{2}{3\hbar\sqrt{\frac{e}{m_e}(eF^2 - \mathbf{v}_\perp \cdot \dot{\mathbf{F}})}}\left[2I_p + \frac{1}{m_e}v_\perp^2 + \frac{1}{m_e}\left(p_z - \left(\frac{p_\perp^2 - v_\perp^2}{2m_e c} + \frac{2m_e I_p + v_\perp^2}{6m_e c}\right)\right)^2\right]^{3/2}\right\}, \tag{14}$$

where $\mathbf{v}_\perp = \mathbf{p}_\perp + e\mathbf{A}(t)$. Clearly, the iso-transverse-energy-$E_\perp$ expectation value of the linear momentum transfer can be obtained as

$$\langle p_z\rangle_\perp = \frac{p_\perp^2 - v_\perp^2}{2m_e c} + \frac{2m_e I_p + v_\perp^2}{6m_e c}. \tag{15}$$

In the adiabatic limit where the average initial transverse momentum $\langle v_\perp\rangle = 0$, we have

$$\langle p_z\rangle_\perp = \frac{p_\perp^2}{2m_e c} + \frac{I_p}{3c} = \frac{E_\perp}{c} + \frac{I_p}{3c}. \tag{2}$$

Now, we aim to obtain the iso-dressed-energy-$E_\gamma$ expectation value of the linear momentum transfer. Along a specific ATI ring, $n\hbar\omega = E_\gamma + I_p + U_p$ is a constant, Eq. (14) can thereby be rewritten as

$$W_{\text{ndSPA}}(\mathbf{p}) = \exp\left\{-\frac{2\left[2n\hbar\omega + 2\left(1 + \frac{p_z}{m_e c}\right)\frac{e}{m_e}\mathbf{p} \cdot \mathbf{A}\right]^{3/2}}{3\hbar\sqrt{\left(1 + \frac{p_z}{m_e c}\right)\frac{e}{m_e}(eF^2 - \mathbf{v}_\perp \cdot \dot{\mathbf{F}})}}\right\}$$
$$= \exp\left\{-\frac{2}{3\hbar\omega}\frac{\left[2n\hbar\omega - 2\left(1 + \frac{p_z}{m_e c}\right)\frac{e}{m_e}p_\perp A_0\right]^{3/2}}{\left[\left(1 + \frac{p_z}{m_e c}\right)\frac{e}{m_e}p_\perp A_0\right]^{1/2}}\right\} \tag{16}$$

for long circular pulses. This ionization rate maximizes when $[1 + p_z/(m_e c)]p_\perp$ peaks, leading to

$$\langle\cos\theta_\gamma\rangle_\gamma = \frac{\sqrt{2m_e E_\gamma}}{m_e c}. \tag{17}$$

The iso-dressed-energy-$E_\gamma$ expectation value of the linear momentum transfer is thereby

$$\langle p_z\rangle_\gamma = \sqrt{2m_e E_\gamma}\langle\cos\theta_\gamma\rangle_\gamma - \frac{U_p}{c} = 2\frac{E_\gamma}{c} - \frac{U_p}{c}. \tag{4}$$

**Nondipole Keldysh–Faisal–Reiss theory**

The SFA is often used as a synonym to the Keldysh–Faisal–Reiss theory (KFR)[43-45] in the literature, although there are some minor differences[46]. To distinguish our theoretical approaches in this work, we term the approach with time-domain integrations as SFA, such as in Eq. (13), and name the approach with energy-domain summations as KFR, such as in Eq. (18).

In order to gain further insights into the physical origin of the photon-number-dependent linear momentum transfer, we extend the dipole version of KFR[47] by applying the long-pulse approximation to ndSFA in the velocity gauge, leading to the ndKFR model for circular pulses

$$M_{\text{ndKFR}}(\mathbf{p}) = \left(1 - \frac{p_z}{m_e c}\right)\left(E + I_p\right)\psi_0\left(\mathbf{p} - \frac{E + I_p}{c}\hat{\mathbf{e}}_z\right)$$
$$2\pi i\sum_n J_n(\alpha)e^{in\phi_\mathbf{p}}\delta\left(E_\gamma + I_p + U_p - n\hbar\omega\right) \tag{18}$$

in a photon-number-resolved manner, where $\tan\phi_{\mathbf{p}} = p_y/p_x$, $\alpha = -eA_0 p_\perp [1 + p_z/(m_e c)]/(m_e \hbar\omega)$ is a dimensionless parameter, and $J_n$ is the Bessel function. Clearly, the Dirac delta function in Eq. (18) gives rise to the ATI peaks. Along these iso-dressed-energy-$E_\gamma$ surfaces, on the other hand, the prefactor $J_n(\alpha)$ reshapes how the probability distributes.

Defining $p_\gamma$ as the dressed radial distance, $\theta_\gamma$ the dressed polar angle and $\phi$ the azimuthal angle with respect to $(0, 0, -U_p/c)$, the expectation value of photon-number-resolved linear momentum transfer is expressed by

$$\langle p_z \rangle_\gamma = \frac{\int d\theta_\gamma d\phi\, p_\gamma^3 \sin\theta_\gamma \cos\theta_\gamma |M_{\text{ndKFR}}(\mathbf{p})|^2}{\int d\theta_\gamma d\phi\, p_\gamma^2 \sin\theta_\gamma |M_{\text{ndKFR}}(\mathbf{p})|^2} - \frac{U_p}{c}. \quad (19)$$

We note that the asymptotic part of the bound-state wave function dominates the tunneling ionization, whose Fourier transform is

$$\psi_0(\mathbf{k}) \simeq \left(\kappa^2 \hbar^2\right)^{\nu+\frac{1}{4}} \frac{1}{\left(\kappa^2 \hbar^2 + k^2\right)^{\nu+1}} \quad (20)$$

with $\kappa = \sqrt{2 m_e I_p}/\hbar$. In addition, $n > |\alpha|$ in the present study, which justifies the approximation[45]

$$J_n(\alpha) \approx \frac{1}{n!}\left(\frac{1}{2}\alpha\right)^n. \quad (21)$$

A direct evaluation of Eq. (19) gives the photon-number-resolved linear momentum transfer

$$\langle p_z \rangle_\gamma = \frac{4\left(n + \frac{\nu}{1 - \frac{U_p}{n\hbar\omega}}\right)}{2n+3} \frac{E_\gamma}{c} - \frac{U_p}{c}. \quad (5)$$

The angular momentum of the initial state contributes little to Eq. (5) for multi-photon ionization; as we have seen above, it is the exponent part, which is universal, that determines the momentum transfer.

## Data availability
The data used in this study are available in the Zenodo database https://doi.org/10.5281/zenodo.15584815.

## Code availability
The codes used in this study are available in the Zenodo database https://doi.org/10.5281/zenodo.15584815.

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

## Acknowledgements

This work was supported by the Innovation Program for Quantum Science and Technology (Grant No. 2024ZD0300700 J.W.), the National Natural Science Foundation of China (Grants No. 12474341 N.H., No. 12227807 J.W., No. 12241407 J.W., No. 11925405 F.H., and No. 12474348 K.L.), the Science and Technology Commission of Shanghai Municipality (Grant No. 23JC1402000 J.W.), and the Shanghai Pilot Program for Basic Research (Grant No. TQ20240204 J.W.). The experimental work was funded by DFG. K.L. acknowledges support by the Startup Funding of Zhejiang University. Numerical computations were in part performed on the ECNU Multifunctional Platform for Innovation (001).

## Author contributions

X.M., H.N., P.H., H.L. and F.H. performed theoretical predictions and numerical simulations, K.L., S.E., K.U., R.D. and J.W. conducted experimental verifications, all authors participated in the discussions and contributed to the writing of the manuscript.

## Competing interests

The authors declare no competing interests.
