## [Transparent Peer Review file · Nature Communications]

Photon Momentum Transfer and Partitioning: From One to Many

Corresponding Author: Dr Hongcheng Ni

Version 0:

Reviewer comments:

Reviewer #1

(Remarks to the Author)

Under the title “Photon Momentum Transfer and Partitioning: From One to Many” Mao et al. present a combination of experimental and theoretical study on the transfer of linear momentum from the absorbed photons to the emitted photoelectron in the strong field ionization process. The authors introduce the concept of dressed energy iso surfaces, which allows them to disentangle the photoelectron momentum distribution recorded from such ionization events in a meaningful manner in contrast to most of the earlier experimental studies in the field. The experimental results are in good agreement with the theoretical (analytical) description of the linear momentum transfer that the authors develop in this paper. Despite the experimentally challenging study I see the main novelty in a simple mathematical trick to sort the ensemble of the photoelectrons according to their physical origin (number of absorbed photons during the ionization process) which leaves this study with limited novelty since similar methods have been applied in literature (see e.g. elliptical coordinates in the analysis of attoclock/angular streaking experiments). Nevertheless, the findings of the study are of high importance for the community and therefore I suggest the work for publication in Nature Communications after revisions. In general, the manuscript is well written, the findings are presented clearly and the conclusions drawn from the experimental data are solid. Nevertheless, to strengthen the manuscript further I would like the authors to address my following concerns prior publication.

1: The wording in lines 79 – 80 is slightly misleading. The measurements do not use the “standing wave in the vacuum chamber of the COLTRIMS reaction microscope”. For the presented results, one of the laser beam directions is blocked before the interaction point. Surely, the standing wave configuration would be an appealing addition to the experimental study since here the transfer of linear momentum shall cancel (vanishing shift of the photoelectron momentum distribution in laser beam direction (p_z)). Is such data available and could be included at least in the supplementary information as a proof for no systematic errors in the experimental data?

2: Please comment on the quality of circular polarization, what is the residual ellipticity on the forward and backward propagating beams in the reaction point of the COLTRIMS?

3: Why is the data presented in Figure 2 not shown for the two separate experimental conditions (forward direction and backward light propagation direction). Does this point at a systematic error in the measurement that is covered up here by the presentation? It would be important to show the corresponding data at least in the supplementary.

4: Please comment on the magnetic field strength and homogeneity used in the COLTRIMS to guide the electrons and ions to their respective detector surfaces. What is the error introduced onto the momentum precision by imperfections of the homogeneity of the magnetic field in the drift regions of the COLTRIMS?

5: Please comment if the recorded events in the COLTRIMS have been filtered on coincidence detection, i.e. exactly one Xe-Ion and one electron or have more relaxed filters been applied to increase the statistics?

6: Which integration range / binning in iso-dressed-energy- E_γ direction for experimental points (Figure 2) has been applied. To my understanding it would make sense to choose the bins around the ATI peaks, which seem to be in contradiction to the finer binning of the presented data.

7: Figure 2 main text and S2: What is included in the error bars for the experimental data. Is this just the uncertainty from the gaussian fit to extract the peak of the distribution or are systematic uncertainties for the spectrometer calibration included?

8: It would be very interesting to perform the experiment at different wavelengths to strengthen the result to check the suggested scaling laws also as a function of photon energy. Possible wavelength would be 400 nm and longer wavelength towards the mid-IR, where it is known that the importance of the linear momentum transfer is elevated). Especially 400 nm

should be easily accessible via frequency doubling of the laser system at hand. If possible, the authors shall include this experimental data in the manuscript or supplementary information.

Reviewer #2

(Remarks to the Author)

The manuscript my Mao and colleagues points to the role of photon momentum in photoemission momentum microscopy. They instigate pure photoemission by using circularly polarized optical pulse and analyze the momentum components that are parallel to the photon momentum, p_z , separately from the perpendicular momentum, p_{\perp} . Their main point is that the center momentum-space probability density is not at the zero-momentum point, but rather shifted on the p_z -axis. Hence, they argue that previous work that relied on integrating on equi-energy spheres, marked as $\langle p_z \rangle_{\perp}$ are wrong, or should be regarded with care. They show that when integrating with respect to the more exact center, $\langle p_z \rangle_{\gamma}$, provides a factor of 2. The authors explain this factor in an intuitive manner, by accounting for the momentum exchange between the absorbed photon, electron, and parent ion.

This is an interesting paper, and I would support publication, considering the following minor comments:

Please clarify if a.u. in the figure 2 is atomic units or arbitrary units.

Please elaborate more on the role of U_p . The explanation suggests that it is a given, fixed, energy that goes back to the field. But is it? U_p is typically an averaged quantity for a free electron quiver in an EM field.

The trace in Fig. 2 is consistently lower than the predicted curve. Could you comment on the physical origin of that shift?

Reviewer #3

(Remarks to the Author)

Due to the many equation in my review is is submitted as a pdf-file below

Version 1:

Reviewer comments:

Reviewer #2

(Remarks to the Author)

I have reviewed the point-by-point response of the authors to my comments and to the comments of Reviewer #1, and found them detailed and satisfactory.

The authors corrected and clarified in the issues that were raised in a professional manner.

Therefore, I support the publication of the manuscript in its current form.

Reviewer #3

(Remarks to the Author)

I have read the revised manuscript and the answers to all the three referees. Concerning my previous report I am happy that the authors now have fixed the "inconsistencies in the equations" in the manuscripts. But regarding the discussion they provide in the answer, I am astonished that they do not see the difference between using a reduced systems of units in calculations (which of course is the "standard practice in the field") and to remove natural constants from the equations in a manuscript. The latter is sometimes done, but it is by no means standard. Many of the articles cited by the authors use $\hbar\omega$ for the photon energy - instead of just ω . In fact the authors themselves keep c in the equations although its value in atomic units is $1/\alpha$ (α = the fine structure constant $=1/137\dots$). To include \hbar and m_e in the equations, as I did in my report, does not make them more clumsy or less accurate, just more transparent. It is of course up to the authors (and the journal) to decide how reader friendly they want to be, but I note that the sentence "atomic units are used throughout unless stated otherwise" on page 3, is not accompanied by their definition. To include this definition is indeed the "standard practice in the field".

Color legend:

Red: changes made in the manuscript

Blue: excerpts from the Reviewer Reports

Black: our responses to the Reviewer comments

REVIEWER 1

Under the title “Photon Momentum Transfer and Partitioning: From One to Many” Mao et al. present a combination of experimental and theoretical study on the transfer of linear momentum from the absorbed photons to the emitted photoelectron in the strong field ionization process. The authors introduce the concept of dressed energy iso surfaces, which allows them to disentangle the photoelectron momentum distribution recorded from such ionization events in a meaningful manner in contrast to most of the earlier experimental studies in the field. The experimental results are in good agreement with the theoretical (analytical) description of the linear momentum transfer that the authors develop in this paper. Despite the experimentally challenging study I see the main novelty in a simple mathematical trick to sort the ensemble of the photoelectrons according to their physical origin (number of absorbed photons during the ionization process) which leaves this study with limited novelty since similar methods have been applied in literature (see e.g. elliptical coordinates in the analysis of attoclock/angular streaking experiments). Nevertheless, the findings of the study are of high importance for the community and therefore I suggest the work for publication in Nature Communications after revisions. In general, the manuscript is well written, the findings are presented clearly and the conclusions drawn from the experimental data are solid. Nevertheless, to strengthen the manuscript further I would like the authors to address my following concerns prior publication.

We thank the Reviewer for the positive assessment of our work, stating that it is of high importance, well written, clearly presented, and solid.

1: The wording in lines 79 – 80 is slightly misleading. The measurements do not use the “standing wave in the vacuum chamber of the COLTRIMS reaction microscope”. For the presented results, one of the laser beam directions is blocked before the interaction point. Surely, the standing wave configuration would be an appealing addition to the experimental study since here the transfer of linear momentum shall cancel (vanishing shift of the photoelectron momentum distribution in laser beam direction (p_z)). Is such data available and could be included at least in the supplementary information as a proof for no systematic errors in the experimental data?

We appreciate the suggestion of the Reviewer. This question is addressed in conjunction with Q3.

The data for the standing wave (SW) and the forward and backward propagating cases are depicted in Fig. R1. While the nondipole shift from the SW should theoretically cancel out, our experimental results show non-zero values due to the inhomogeneous response of the detector. This underscores the necessity of conducting a SW experiment, where we subtract the “backward curve” from the “forward curve” and then divide by 2 to eliminate systematic errors, as illustrated in Fig. 2 of the manuscript. To further validate the removal of systematic errors by the SW, we divided the data into two subsets based on the electron’s momentum along the polarization direction: one subset with positive p_x and the other with negative p_x . Given the symmetry of the light along the polarization direction, the nondipole curves for these two subsets should be similar, as demonstrated in the data presented in Fig. R1. We assess the validity of our data based on the following criteria:

1. The SW curve does not need to be centered at 0 due to detector inhomogeneity, but it should lie midway between the “forward curve” and “backward curve.”
2. The two subsets corresponding to positive and negative p_x should exhibit similar nondipole curves due to the symmetry along the polarization direction.

As detailed in the manuscript, we alternated between the three cases during data collection. However, to enhance collection efficiency, the collection time for the SW case was much shorter than for the “forward” and “backward” cases. This is because the SW data is primarily used as a cross-check to ensure it lies roughly between the “forward” and “backward” curves. Specifically, the statistics for the “forward” or “backward” cases are 15 times higher than those for the SW case.

The detailed discussion regarding the experimental data for different laser propagation directions is presented as Sec. S4 of the Supplementary Materials. In addition, in order to avoid confusion to the reader, we have added the following sentence in the main text:

During the measurement of linear momentum transfer, one of the two counter-propagating lasers is turned off.

FIG. R1. Linear momentum transfer as a function of the **a** transverse energy E_{\perp} and **b** dressed energy E_{γ} for the forward-propagating (along $+\hat{e}_z$ direction), backward-propagating (along $-\hat{e}_z$ direction) and standing wave (SW) cases. The positive and negative energies represent those electrons with positive and negative momentum along the polarization direction (\hat{e}_x).

2: Please comment on the quality of circular polarization, what is the residual ellipticity on the forward and backward propagating beams in the reaction point of the COLTRIMS?

We appreciate the comment of the Reviewer. To make this clear to the readers, we have added the following text in the manuscript:

To ensure circular polarization of the laser field entering the vacuum chamber, we carefully controlled its polarization state using a combination of a half-wave plate and a quarter-wave plate in both pathways. This setup effectively compensates the differences in reflectivity between the *s*- and *p*-polarization components of the circularly polarized light. The degree of circular polarization achieved was estimated to be better than 0.9, approaching the ideal value of 1.

3: Why is the data presented in Figure 2 not shown for the two separate experimental conditions (forward direction and backward light propagation direction). Does this point at a systematic error in the measurement that is covered up here by the presentation? It would be important to show the corresponding data at least in the supplementary.

See answer to Q1.

4: Please comment on the magnetic field strength and homogeneity used in the COLTRIMS to guide the electrons and ions to their respective detector surfaces. What is the error introduced onto the momentum precision by imperfections of the homogeneity of the magnetic field in the drift regions of the COLTRIMS?

We appreciate the comment of the Reviewer. In our experimental setup, we did not employ a magnetic field to guide electrons and ions. Instead, we utilized a μ -metal shield within the vacuum chamber to effectively screen external magnetic fields, particularly the Earth's magnetic field. While this design was not specifically required for the current experiment, it was implemented to accommodate other experimental requirements. Since the setup was initially designed with this feature, we retained it for consistency and potential future use. To make this clear to the readers, we have added the following text in the manuscript:

In our experimental setup, we did not employ a magnetic field to guide electrons and ions. Instead, we utilized a μ -metal shield within the vacuum chamber to effectively screen external magnetic fields, particularly the Earth's magnetic field.

5: Please comment if the recorded events in the COLTRIMS have been filtered on coincidence detection, i.e. exactly one Xe-Ion and one electron or have more relaxed filters been applied to increase the statistics?

We thank the Reviewer for the question. To clarify, we have added the following text in the manuscript:

We employed coincident detection of the Xe ion (Xe^+) and the electron (e^-). By applying a momentum gate along the polarization direction, we ensured that only the ion and electron pairs originating from the same parent atom were selected,

based on momentum conservation.

6: Which integration range / binning in iso-dressed-energy- E_γ direction for experimental points (Figure 2) has been applied. To my understanding it would make sense to choose the bins around the ATI peaks, which seem to be in contradiction to the finer binning of the presented data.

We appreciate the question of the Reviewer. The binning size used in Figure 2 is 0.02. We have explored the impact of varying the binning size on our results. The findings indicate that the overall trends remain consistent across different binning sizes. The reasons are two fold. First, the conclusions remain valid even in the absence of ATI peaks, such as in the case of tunneling ionization. Second, the stability can be further attributed to the volume or intensity averaging inherent in our experimental approach, which renders the results less sensitive to binning variations than one would assume. To make this clear to the readers, we have added the following text in the manuscript:

The binning size employed is 0.02 a.u., and the results are robust against variations in binning size.

7: Figure 2 main text and S2: What is included in the error bars for the experimental data. Is this just the uncertainty from the gaussian fit to extract the peak of the distribution or are systematic uncertainties for the spectrometer calibration included?

We appreciate the valuable question of the Reviewer. To clarify, we have added the following text in the manuscript:

The error bars in Fig. 2 represent statistical uncertainties associated with the experimental events. Given that our analysis involves the use of a standing wave, and the final results are derived by subtracting the forward and backward curves, any uncertainties related to the spectrometer calibration effectively cancel out along the light-propagation direction. Consequently, the primary source of uncertainty in our results is statistical in nature, stemming from the inherent variability of the experimental measurements.

8: It would be very interesting to perform the experiment at different wavelengths to strengthen the result to check the suggested scaling laws also as a function of photon energy. Possible wavelength would be 400 nm and longer wavelength towards the mid-IR, where it is known that the importance of the linear momentum transfer is elevated). Especially 400 nm should be easily accessible via frequency doubling of the laser system at hand. If possible, the authors shall include this experimental data in the manuscript or supplementary information.

We appreciate the suggestion of the Reviewer. We concur with the Reviewer that exploring different driving wavelengths would be highly valuable. However, implementing this in our current setup poses significant challenges. Our vacuum window is specifically coated with an anti-reflective (AR) coating for 800 nm, rendering it highly reflective (HR) at 400 nm. To accommodate other wavelengths, we would need to replace the vacuum window, which involves procuring new windows, which is a time-consuming process. Additionally, after replacing the window, we would need to bake the chamber to restore high vacuum conditions. This entire process is both time-intensive and laborious. Given these constraints, conducting experiments with different wavelengths using our current setup would be a substantial undertaking. We appreciate your understanding of these limitations.

We hope we have fully addressed the concerns of the Reviewer and our improved manuscript meets the criteria to be published in Nature Communications.

REVIEWER 2

The manuscript my Mao and colleagues points to the role of photon momentum in photoemission momentum microscopy. They instigate pure photoemission by using circularly polarized optical pulse and analyze the momentum components that are parallel to the photon momentum, p_z , separately from the perpendicular momentum, p_\perp . Their main point is that the center momentum-space probability density is not at the zero-momentum point, but rather shifted on the p_z -axis. Hence, they argue that previous work that relied on integrating on equi-energy spheres, marked as \perp are wrong, or should be regarded with care. They show that when integrating with respect to the more exact center, γ , provides a factor of 2. The authors explain this factor in an intuitive manner, by accounting for the momentum exchange between the absorbed photon, electron, and parent ion.

This is an interesting paper, and I would support publication, considering the following minor comments:

We thank the Reviewer for the positive assessment of our work and for recognizing that our study is interesting.

Please clarify if a.u. in the figure 2 is atomic units or arbitrary units.

We appreciate the suggestion of the Reviewer. The a.u. in the figure 2 is atomic units. To make this clear to the readers, we have revised the following text in the manuscript:

[atomic units (a.u.) are used throughout unless stated otherwise]

Please elaborate more on the role of U_p . The explanation suggests that it is a given, fixed, energy that goes back to the field. But is it? U_p is typically an averaged quantity for a free electron quiver in an EM field.

We appreciate the question of the Reviewer. In the manuscript, the ponderomotive energy $U_p = (\int_0^T [\mathbf{A}^2(t)/2]dt)/T$, where T denotes the period of the laser field and $\mathbf{A}(t)$ represents the vector potential, signifies the average quiver energy of a free electron within a laser field over one cycle. For a long circularly polarized laser field, $U_p = A_0^2/2$ remains constant. To make this clear to the readers, we have added the following text in the manuscript:

The ponderomotive energy U_p represents the average quiver energy of a free electron in a laser field. In a long circular laser field, $U_p = A_0^2/2 = F_0^2/2\omega^2$ remains constant, with A_0 and F_0 being the amplitudes of the vector potential and the electric field, respectively.

The trace in Fig. 2 is consistently lower than the predicted curve. Could you comment on the physical origin of that shift?

We appreciate the insightful question of the Reviewer. In Fig. 2a of the main text, the experimental results for $\langle p_z \rangle_{\perp}$ consistently fall below the theoretical curve $E_{\perp}/c + I_p/3c$. This discrepancy arises because the theoretical curve is based on the adiabatic approximation and neglects the influence of the prefactor in the ionization rate derived from nondipole saddle-point approximation (ndSPA). To address this, we have derived the average linear momentum transfer in the full setting, as previously reported in Ref. [1]:

$$\begin{aligned} \langle p_z \rangle_{\perp}^{(\text{NA}, \alpha_Z)} &= \frac{\langle p_{\perp}^2 \rangle}{2c} - \frac{\langle v_{\perp}^2 \rangle}{2c} + \left[1 - \frac{2\alpha_Z F_0}{(2I_p)^{3/2}} \right] \frac{2I_p + \langle v_{\perp}^2 \rangle}{6c} \\ &= \frac{E_{\perp}}{c} - \frac{\langle v_{\perp}^2 \rangle}{2c} + \left[1 - \frac{2\alpha_Z F_0}{(2I_p)^{3/2}} \right] \frac{2I_p + \langle v_{\perp}^2 \rangle}{6c} \\ &= \frac{\Delta E_{\perp}}{c} + \delta \frac{\tilde{I}_p}{3c}, \end{aligned} \quad (\text{R1})$$

where

$$\Delta E_{\perp} = E_{\perp} - E_{\perp 0} = E_{\perp} - \frac{1}{2} \langle v_{\perp}^2 \rangle \quad (\text{R2})$$

denotes the energy absorption during the continuum motion after tunneling,

$$\tilde{I}_p = I_p + E_{\perp 0} = I_p + \frac{1}{2} \langle v_{\perp}^2 \rangle \quad (\text{R3})$$

is the effective ionization potential accounting for the initial kinetic energy at the tunnel exit, and

$$\delta = 1 - \frac{2\alpha_Z F_0}{(2I_p)^{3/2}} = 1 - \frac{2(1+Z/\sqrt{2I_p})F_0}{(2I_p)^{3/2}} \quad (\text{R4})$$

roots in the prefactor in the ndSPA ionization rate. For nonadiabatic tunneling, the influence of the initial kinetic energy $E_{\perp 0}$ at the tunnel exit cannot be neglected [1]:

$$E_{\perp 0} = \frac{1}{2} \langle v_{\perp}^2 \rangle = \frac{1}{2} \langle v_{\perp} \rangle^2 + \frac{F_0}{4\sqrt{2I_p}} = \frac{1}{2} \left(\frac{I_p}{3A_0} \right)^2 + \frac{F_0}{4\sqrt{2I_p}}. \quad (\text{R5})$$

If we assume adiabatic tunneling with $E_{\perp 0} \rightarrow 0$ and neglect the influence of the prefactor by setting $\alpha_Z \rightarrow 0$ (or $\delta \rightarrow 1$), we obtain the commonly used simplified formula:

$$\langle p_z \rangle_{\perp}^{(\text{A}, \alpha_Z=0)} = \frac{E_{\perp}}{c} + \frac{I_p}{3c}. \quad (\text{R6})$$

Fig. R2 illustrates the linear momentum transfer $\langle p_z \rangle_{\perp}$ as a function of the transverse energy E_{\perp} . The black dashed line represents the simplified formula [Eq. (R6)], while the orange dashed line represents the full expression [Eq. (R1)]. The blue

FIG. R2. Linear momentum transfer $\langle p_z \rangle_{\perp}$ as a function of the transverse energy E_{\perp} . The black dashed line represents Eq. (R6), the orange dashed line represents Eq. (R1), and the blue line represents the experimental results.

line corresponds to the experimental results. As evident in Fig. R2, the orange dashed curve, which accounts for nonadiabatic effects and the ndSPA prefactor, lies consistently lower than the black dashed curve. This downward shift better aligns with the experimental results, highlighting the importance of considering nonadiabatic effects and the prefactor in the theoretical model.

To clarify this point, we have revised the following text in the manuscript:

While $\langle p_z \rangle_{\perp}^{(\text{Exp.})}$ shows a qualitative agreement with [9,23]

$$\langle p_z \rangle_{\perp} = \frac{E_{\perp}}{c} + \frac{I_p}{3c}, \quad (\text{R7})$$

or a better agreement with [20]

$$\langle p_z \rangle_{\perp} = \frac{\Delta E_{\perp}}{c} + \delta \frac{\tilde{I}_p}{3c} = \frac{E_{\perp} - E_{\perp 0}}{c} + \delta \frac{I_p + E_{\perp 0}}{3c}, \quad (\text{R8})$$

where $\langle \cdot \rangle_{\perp}$ denotes integration along the iso-transverse-energy- E_{\perp} surface, $E_{\perp 0}$ denotes the initial kinetic energy at the tunnel exit, \tilde{I}_p represents the effective ionization potential, and δ originates from the prefactor in the ionization rate (see Supplementary Materials for details)...

In addition, we have incorporated the detailed discussion as Sec. S3 in the Supplementary Materials, addressing the reasons behind the overall deviation of the experimental results below the simplified theoretical curve [Eq. (R7)] in Fig. 2a.

We hope we have fully addressed the concerns of the Reviewer and our improved manuscript meets the criteria to be published in Nature Communications.

REVIEWER 3

This manuscript addresses interesting questions: the transfer of linear momentum in (single) photoionization, caused by an arbitrarily number of photons, and the momentum partitioning between the electron and the parent ion. They present truly new findings, convincingly demonstrated by experimental results in Fig.2.

We thank the Reviewer for the positive assessment of our work, stating that our work is interesting, truly new, and convincing.

The argumentation is, however, quite hard to follow. Many steps in the derivations are left to the reader and some confusing results are not even commented on. The labelling is also inconsistent. I describe my confusion below and hope that the authors can straighten up the explanation.

We appreciate the criticism of the Reviewer. In the following, we address the concerns point-by-point.

In the following discussion I have chosen to use dimensionally correct expressions. Something the authors have chosen not to do. While it is obvious that one has to choose a particular unit system when giving quantitative results, it is a mystery to me why anybody wants to give equations in “atomic units”. This habit to drop for example the electron mass or \hbar from equations is perhaps not a serious problem, but a rather annoying one, and it makes it harder to see through the equations.

Using atomic units in scientific research, particularly in the fields of atomic, molecular, and optical (AMO) physics, offers several significant benefits:

1. Simplification of Equations: Atomic units simplify the form of fundamental equations by setting key constants to unity. This simplification eliminates the need to carry these constants through calculations, making equations more concise and easier to handle.
2. Natural Scaling for Atomic Systems: Atomic units are naturally scaled to the size and energy scales typical of atomic systems. For example, lengths are measured in units of the Bohr radius a_0 , energies are measured in Hartrees E_h , and time is measured in units of $\hbar/E_h \approx 24$ as. These units are particularly well-suited for describing phenomena at the atomic scale, making it easier to interpret results and compare them with experimental data.
3. Enhanced Intuition: Using atomic units helps build an intuitive understanding of atomic-scale phenomena. For example, an electron in the ground state of a hydrogen atom has an energy of -0.5 Hartrees and an orbital radius of 1 Bohr radius. This intuitive scaling makes it easier to grasp the relative magnitudes of various quantities and their physical significance.
4. Reduction of Numerical Errors: By working in atomic units, numerical calculations often involve numbers of similar magnitude, reducing the risk of numerical errors associated with very large or very small values. This is particularly important in high-precision calculations and simulations.

Given these benefits of atomic units, their use has become a standard practice in the field. Accordingly, we have employed atomic units throughout the present study.

The authors assume laser propagation in the z -direction and start with the the beyond dipole corrections to the ponderomotive potential for an electron (the expression for U_p is never given, i.e. it is assumed that the reader knows it by heart)

$$U'_p = \left(1 + \frac{p_z}{m_e c}\right) U_p. \quad (\text{R9})$$

We thank the Reviewer for pointing out this issue. As suggested by the Reviewer, we have added an explanation of U_p in the manuscript:

The ponderomotive energy U_p represents the average quiver energy of a free electron in a laser field. In a long circular laser field, $U_p = A_0^2/2 = F_0^2/2\omega^2$ remains constant, with A_0 and F_0 being the amplitudes of the vector potential and the electric field, respectively.

Then the kinetic energy of the parent ion is neglected, i.e. it is assumed that all the photon energy is absorbed by the photoelectron. The latter's kinetic energy is thus given as

$$\frac{p^2}{2m_e} = \frac{p_{\perp}^2 + p_z^2}{2m_e} = n\hbar\omega - I_p - U'_p = n\hbar\omega - I_p - U_p - \frac{p_z}{m_e c} U_p \quad (\text{R10})$$

Now the last term is moved to the left-hand side giving

$$\frac{p_{\perp}^2 + p_z^2}{2m_e} + \frac{p_z}{m_e c} U_p = n\hbar\omega - I_p - U_p. \quad (\text{R11})$$

Then there is an additional approximation:

$$\left(p_z + \frac{U_p}{c}\right)^2 = p_z^2 + 2p_z \frac{U_p}{c} + \left(\frac{U_p}{c}\right)^2 \approx p_z^2 + 2p_z \frac{U_p}{c} \quad (\text{R12})$$

When this approximate expression is inserted in the equation above the quantity called E_γ (Eq. 1) is obtained:

$$E_\gamma = \frac{1}{2m_e} \left(p_\perp^2 + \left(p_z + \frac{U_p}{c} \right)^2 \right) = n\hbar\omega - I_p - U_p \quad (\text{R13})$$

Note that it is only this last equation that is given in the manuscript.

Concerning the momentum partitioning it is found that the momentum transferred to the electron is

$$p_e = 2\frac{E_\gamma}{c} - \frac{U_p}{c} \quad (\text{R14})$$

while that to the ion is

$$p_i = -\frac{E_\gamma}{c} + \frac{I_p}{c} \quad (\text{R15})$$

giving a total momentum transfer to the electron-ion system of

$$p_e + p_i = \frac{E_\gamma}{c} - \frac{U_p}{c} + \frac{I_p}{c} = \frac{n\hbar\omega}{c} - 2\frac{U_p}{c}. \quad (\text{R16})$$

I would have liked to get some help with the interpretation of this result. Why isn't the sum of the two particles momentum equal to the absorbed photon momentum?

We are grateful to the Reviewer for the careful review of our manuscript and for identifying the inconsistencies. As pointed out by the Reviewer, the two-stage photoionization model could lead to confusions to the readers, and thus we have omitted this model. Instead, we elucidate the rationale behind the usage of the dressed energy E_γ , after which we present directly how the photon momentum is partitioned between the photoelectron and the ion. Accordingly, we have made the following revisions to the text:

Field dressing and photon momentum partitioning

At last, we would like to elucidate the rationale behind employing the ‘‘dressed energy’’ E_γ as a means to quantify the partitioning of photon momentum transfer. Eq. (1) suggests that both the photoelectron energy and linear momentum in the ATI process are simultaneously influenced by the laser dressing, manifested as the U_p term. At low laser intensity, U_p in Eq. (1) is nearly negligible, and the photoionization process is mainly influenced by the frequency of the laser field. In contrast, under a strong laser field, the photoelectron returns an energy U_p to the field, with a concurrent return of linear momentum U_p/c to the field. The energy return of U_p is evident in the last term of Eq. (1), while the linear momentum return of U_p/c is observed as a common shift to the center of all ATI momentum rings. Not surprisingly, these energy and linear momentum exchanges adhere to the photon's dispersion relation. Within the dipole approximation, it is valid to consider only the influence of the laser field on the energy. However, in the nondipole regime, we must simultaneously analyze the consistent adjustments to both the photoelectron energy and linear momentum. Therefore, it is helpful to define a ‘‘dressed energy’’ E_γ as given in Eq. (1), which represents the center-shifted full photoelectron energy. The virtue of E_γ is that, in the nondipole case, this quantity is quantized analogously to the the quantization of the photoelectron energy in the dipole case.

We consider now the partitioning of the linear momentum between the photoelectron and the ion. Under the influence of the laser field, a total energy of $n\omega - U_p = E_\gamma + I_p$ is deposited to the target, and a corresponding linear momentum of $(n\omega - U_p)/c = (E_\gamma + I_p)/c$ is transferred to the center of mass of the photoelectron and the ion. Both experimental and theoretical analyses have verified that an average linear momentum of $2E_\gamma/c - U_p/c$ is transferred to the photoelectron. Thus, by momentum conservation, the residual linear momentum of $-E_\gamma/c + (I_p + U_p)/c$ is transferred to the ion, as illustrated in Fig. 1.

We would like to emphasize that the total linear momentum transferred is not $n\hbar\omega/c$, but rather $n\hbar\omega/c - U_p/c$. This is because, under the influence of the laser field, the photoelectron returns a linear momentum of U_p/c to the field.

It might be that the explanation is thought to be given in Sec. *Two stages of photoionization*, but the information there is not only coming somewhat late, it is also confusing. It is clearly said that in the first photoionization stage a total momentum of

$n\hbar\omega/c$ is transferred to the center of mass of the electron-ion system. This is reassuring. But when exact expressions for the momentum transfer is given the quantity E_γ seems to have a different definition than before: it is stated that $n\hbar\omega = E_\gamma + I_p$, i.e. that $E_\gamma = n\hbar\omega - I_p$, suggesting another definition of E_γ than in Eq. (1). According to the manuscript, the momentum transfer to the electron is $2E_\gamma/c$ and to the ion $-E_\gamma/c + I_p/c$. Assuming the definition $E_\gamma = n\hbar\omega - I_p$ given in Sec. *Two stages of photoionization* we get

$$2\frac{E_\gamma}{c} = \frac{2n\hbar\omega}{c} - \frac{2I_p}{c} \quad (\text{R17})$$

transferred to the electron and

$$-\frac{E_\gamma}{c} + I_p/c = -\frac{n\hbar\omega}{c} + \frac{2I_p}{c} \quad (\text{R18})$$

to the ion, adding up to the expected sum. If now the second stage is that the photoelectron returns U_p/c to the field (i.e. lower its own momentum with this amount) as it is written in the first paragraph of the section (not $-U_p/c$ as it is written in the second part), the final electron momentum is

$$p_e^f = \frac{2n\hbar\omega}{c} - \frac{2I_p}{c} - \frac{U_p}{c} = 2\frac{(n\hbar\omega - I_p)}{c} - \frac{U_p}{c}. \quad (\text{R19})$$

This agrees with the expression in Eq. (3), but of course only if $E_\gamma = n\hbar\omega - I_p$.

Alternatively one can assume that it is the original definition in Eq. (1) that should be used, i.e. $E_\gamma = n\hbar\omega - I_p - U_p$ then the momentum transferred to the electron would be

$$2E_\gamma/c = 2\frac{n\hbar\omega}{c} - 2\frac{I_p}{c} - 2\frac{U_p}{c} \quad (\text{R20})$$

and to the ion

$$-\frac{E_\gamma}{c} + \frac{I_p}{c} = -\frac{n\hbar\omega}{c} + 2\frac{I_p}{c} + \frac{U_p}{c} \quad (\text{R21})$$

with the sum $n\hbar\omega - U_p/c$, which can hardly be right. Still, if the electron now gives U_p/c back to the field the final result for its change of momentum would be

$$p_e^f = 2\frac{n\hbar\omega - I_p - U_p}{c} - \frac{U_p}{c} \quad (\text{R22})$$

which also agrees with Eq. (3), but of course now with E_γ defined as in Eq. 1.

I hope the authors can clarify the definition of E_γ and the issue of the conservation of linear momentum, as well as making the explanation clearer and more consistent.

We are grateful to the Reviewer for the thorough examination of our manuscript and for identifying the inconsistencies. To address these concerns, we remove the misleading expression $E_\gamma = n\omega - I_p$. Now, $E_\gamma = n\omega - I_p - U_p$ is consistently defined as in Eq. (1) of the manuscript. Under this framework, the electron eventually acquires an average linear momentum of

$$p_e = 2\frac{E_\gamma}{c} - \frac{U_p}{c}, \quad (\text{R23})$$

while the ion acquires an average linear momentum of

$$p_i = -\frac{E_\gamma}{c} + \frac{I_p + U_p}{c}. \quad (\text{R24})$$

The sum of their momenta is

$$p_e + p_i = \frac{E_\gamma}{c} + \frac{I_p}{c} = \frac{n\omega}{c} - \frac{U_p}{c}. \quad (\text{R25})$$

I have also a few other comments.

1) The sentence: ‘‘Figure 2 displays the measured photon momentum transferred to the photoelectron for the Xe atom as a function of the transverse energy $\langle p_z \rangle_\perp$ in panel (a) and as a function of the dressed energy $\langle p_z \rangle_\gamma$ in panel (b).’’.

Is better written as: "Figure 2 displays the measured photon momentum transferred to the photoelectron, $\langle p_z \rangle_\perp$, for the Xe atom as a function of the transverse energy E_\perp in panel (a) and $\langle p_z \rangle_\gamma$ as a function of the dressed energy E_γ in panel (b)."

We thank the Reviewer for pointing this out. We have revised the manuscript accordingly to ensure clarity and accuracy.

2) In Eq. 4 the quantity ν has to be dimensionless. Now it looks as it has dimension $1/\sqrt{(\text{Energy})}$. I guess that ν actually is the square root of the ratio between the ionization potential of the hydrogen-like ground state and the ionization potential of the system in question, where the former should be calculated with the asymptotic charge. In this more transparent form the confusing factor of 2 is not needed, and it is transparent for the reader that with $n = 1$ and $U_p = 0$ the result is $8/5$ times the photoelectron kinetic energy divided with the speed of light.

We thank the Reviewer for pointing this out. The Reviewer is indeed correct that $\nu = Z/\sqrt{2I_p/[2I_p^{(H)}]}$ [2] is a dimensionless quantity, where

$$I_p^{(H)} = \frac{1}{2} \frac{m_e e^4}{(4\pi\epsilon_0)^2 \hbar^2} \quad (\text{R26})$$

corresponds to the ionization energy of the ground state of the hydrogen atom. In atomic units, $\hbar = m_e = e = 4\pi\epsilon_0 = 1$ a.u., and thus $2I_p^{(H)} = 1$ a.u., simplifying the form of theoretical derivation. The representation of ν follows that in Refs. [3–5]. We have added a clarification of the dimension of ν in the manuscript:

where the dimensionless parameter $\nu = Z/\sqrt{2I_p/[2I_p^{(H)}]} = Z/\sqrt{2I_p}$, with the ionization energy $I_p = 0.4457$ a.u., the ionization energy of the hydrogen atom $I_p^{(H)} = 0.5$ a.u., and the asymptotic charge $Z = 1$ for the valence shell of Xe.

In conclusion I support publication of the manuscript if the authors can provide a consistent discussion without confusing labelling and conflicting argumentation.

We thank the Reviewer for the positive recommendation of our work to be published in Nature Communications. We hope we have fully addressed the concerns of the Reviewer and our improved manuscript meets the criteria to be published in Nature Communications.

-
- [1] H. Ni, S. Brennecke, X. Gao, P.-L. He, S. Donsa, I. Březinová, F. He, J. Wu, M. Lein, X.-M. Tong, and J. Burgdörfer, Theory of subcycle linear momentum transfer in strong-field tunneling ionization, *Phys. Rev. Lett.* **125**, 073202 (2020).
- [2] A. Perelomov, V. Popov, and M. Terent'ev, Ionization of atoms in an alternating electric field, *Sov. Phys. JETP* **23**, 924 (1966).
- [3] G. Gribakin and M. Y. Kuchiev, Multiphoton detachment of electrons from negative ions, *Phys. Rev. A* **55**, 3760 (1997).
- [4] A. Jašarević, E. Hasović, R. Kopold, W. Becker, and D. Milošević, Application of the saddle-point method to strong-laser-field ionization, *J. Phys. A* **53**, 125201 (2020).
- [5] P.-L. He, M. Klaiber, K. Z. Hatsagortsyan, and C. H. Keitel, Nondipole coulomb sub-barrier ionization dynamics and photon momentum sharing, *Phys. Rev. A* **105**, L031102 (2022).

This manuscript addresses interesting questions: the transfer of linear momentum in (single) photoionization, caused by an arbitrarily number of photons, and the momentum partitioning between the electron and the parent ion. They present truly new findings, convincingly demonstrated by experimental results in Fig.2.

The argumentation is, however, quite hard to follow. Many steps in the derivations are left to the reader and some confusing results are not even commented on. The labelling is also inconsistent. I describe my confusion below and hope that the authors can straighten up the explanation.

In the following discussion I have chosen to use dimensionally correct expressions. Something the authors have chosen not to do. While it is obvious that one has to choose a particular unit system when giving quantitative results, it is a mystery to me why anybody wants to give equations in “atomic units”. This habit to drop for example the electron mass or \hbar from equations is perhaps not a serious problem, but a rather annoying one, and it makes it harder to see through the equations.

The authors assume laser propagation in the z -direction and start with the the beyond dipole corrections to the ponderomotive potential for an electron (the expression for U_p is never given, i.e. it is assumed that the reader knows it by heart)

$$U'_p = \left(1 + \frac{p_z}{m_e c}\right) U_p.$$

Then the kinetic energy of the parent ion is neglected, i.e. it is assumed that all the photon energy is absorbed by the photoelectron. The latter’s kinetic energy is thus given as

$$\frac{\mathbf{p}^2}{2m_e} = \frac{p_{\perp}^2 + p_z^2}{2m_e} = n\hbar\omega - I_p - U'_p = n\hbar\omega - I_p - U_p - \frac{p_z}{m_e c} U_p$$

Now the last term is moved to the left-hand side giving

$$\frac{p_{\perp}^2 + p_z^2}{2m_e} + \frac{p_z}{m_e c} U_p = n\hbar\omega - I_p - U_p.$$

Then there is an additional approximation:

$$\left(p_z + \frac{U_p}{c}\right)^2 = p_z^2 + 2p_z \frac{U_p}{c} + \left(\frac{U_p}{c}\right)^2 \approx p_z^2 + 2p_z \frac{U_p}{c}$$

When this approximate expression is inserted in the equation above the quantity called E_{γ} (Eq. 1) is obtained:

$$E_{\gamma} = \frac{1}{2m_e} \left(p_{\perp}^2 + \left(p_z + \frac{U_p}{c} \right)^2 \right) = n\hbar\omega - I_p - U_p.$$

Note that it is only this last equation that is given in the manuscript.

Concerning the momentum partitioning it is found that the momentum transferred to the electron is

$$p_e = 2 \frac{E_{\gamma}}{c} - \frac{U_p}{c}$$

while that to the ion is

$$p_i = -\frac{E_\gamma}{c} + \frac{I_p}{c}$$

giving a total momentum transfer to the electron-ion system of

$$p_e + p_i = \frac{E_\gamma}{c} - \frac{U_p}{c} + \frac{I_p}{c} = \frac{n\hbar\omega}{c} - 2\frac{U_p}{c}.$$

I would have liked to get some help with the interpretation of this result. Why isn't the sum of the two particles momentum equal to the absorbed photon momentum?

It might be that the explanation is thought to be given in Sec. *Two stages of photoionization*, but the information there is not only coming somewhat late, it is also confusing. It is clearly said that in the first photoionization stage a total momentum of $n\hbar\omega/c$ is transferred to the center of mass of the electron-ion system. This is reassuring. But when exact expressions for the momentum transfer is given the quantity E_γ seems to have a different definition than before: it is stated that $n\hbar\omega = E_\gamma + I_p$, i.e. that $E_\gamma = n\hbar\omega - I_p$, suggesting another definition of E_γ than in Eq. (1). According to the manuscript, the momentum transfer to the electron is $2E_\gamma/c$ and to the ion $-E_\gamma/c + I_p/c$. Assuming the definition $E_\gamma = n\hbar\omega - I_p$ given in Sec. *Two stages of photoionization* we get

$$2\frac{E_\gamma}{c} = \frac{2n\hbar\omega}{c} - \frac{2I_p}{c}$$

transferred to the electron and

$$-\frac{E_\gamma}{c} + I_p/c = -\frac{n\hbar\omega}{c} + \frac{2I_p}{c}$$

to the ion, adding up to the expected sum. If now the second stage is that the photoelectron returns U_p/c to the field (i.e. lower its own momentum with this amount) as it is written in the first paragraph of the section (not $-U_p/c$ as it is written in the second part), the final electron momentum is

$$p_e^f = \frac{2n\hbar\omega}{c} - \frac{2I_p}{c} - \frac{U_p}{c} = 2\frac{(n\hbar\omega - I_p)}{c} - \frac{U_p}{c}.$$

This agrees with the expression in Eq. (3), but of course only if $E_\gamma = n\hbar\omega - I_p$.

Alternatively one can assume that it is the original definition in Eq. (1) that should be used, i.e. $E_\gamma = n\hbar\omega - I_p - U_p$ then the momentum transferred to the electron would be

$$2E_\gamma/c = 2\frac{n\hbar\omega}{c} - 2\frac{I_p}{c} - 2\frac{U_p}{c}$$

and to the ion

$$-\frac{E_\gamma}{c} + \frac{I_p}{c} = -\frac{n\hbar\omega}{c} + 2\frac{I_p}{c} + \frac{U_p}{c}$$

with the sum $n\hbar\omega/c - U_p/c$, which can hardly be right. Still, if the electron now gives U_p/c back to the field the final result for its change of momentum would be

$$p_e^f = 2\frac{(n\hbar\omega - I_p - U_p)}{c} - \frac{U_p}{c}$$

which also agrees with Eq. (3), but of course now with E_γ defined as in Eq. 1.

I hope the authors can clarify the definition of E_γ and the issue of the conservation of linear momentum, as well as making the explanation clearer and more consistent.

I have also a few other comments.

1) The sentence: “Figure 2 displays the measured photon momentum transferred to the photoelectron for the Xe atom as a function of the transverse energy $\langle p_z \rangle_\perp$ in panel (a) and as a function of the dressed energy $\langle p_z \rangle_\gamma$ in panel (b).”

Is better written as: “Figure 2 displays the measured photon momentum transferred to the photoelectron, $\langle p_z \rangle_\perp$, for the Xe atom as a function of the transverse energy E_\perp in panel (a) and $\langle p_z \rangle_\gamma$ as a function of the dressed energy E_γ in panel (b).”

2) In Eq. 4 the quantity ν has to be dimensionless. Now it looks as it has dimension $1/\sqrt{\text{Energy}}$. I guess that ν actually is the square root of the ratio between the ionization potential of the hydrogen-like ground state and the ionization potential of the system in question, where the former should be calculated with the asymptotic charge. In this more transparent form the confusing factor of 2 is not needed, and it is transparent for the reader that with $n = 1$ and $U_p = 0$ the result is $8/5$ times the photoelectron kinetic energy divided with the speed of light.

In conclusion I support publication of the manuscript if the authors can provide a consistent discussion without confusing labelling and conflicting argumentation.